# Methods for Calculation of Stormwater Treatment Required for Meeting Receiving Waters Quality: Application in a Case Study

## Thomas Larm * and Anna Wahlsten

StormTac Corporation, SE-116 40 Stockholm, Sweden
* Correspondence: thomas.larm@stormtac.com

**Abstract:** There is a need for tools relating stormwater effluent quality data to the ambient water quality guidelines listed, e.g., in the EU Water Framework Directive. We conducted a literature study and identified four methods to calculate the annual required treatment loads (kg/year) to satisfy the guidelines: (1) StormTac Web, (2) the Vollenweider OECD load model, (3) the model developed by the Swedish Water Authorities and SMHI, and (4) the method using effluent concentration criteria. In this paper, we compile and derive new equations based on these methods to enable calculation of annual acceptable load (kg/year) and annual required treatment load (kg/year) for different types of receiving waters and substances, i.e., not only nutrients but also priority chemicals listed in the Water Framework Directive. Obligatory input data consist of calculated or measured annual flow (runoff and baseflow), actual concentrations and criteria concentrations in the receiving water, or criteria concentration for discharge to the receiving waters. The four methods were applied to address total phosphorus in the urban watercourse of Bällstaån, Sweden. Results showed agreement of the calculated required treatment removals with measured phosphorus concentrations in the receiving water when Methods 1–2 were used but more varied results for Methods 3–4, which neglect the receiving waters concentrations and criteria. Based on these results, we recommend Methods 1–2 for use and further evaluation in projects related to improvements in the quality of receiving waters, as they are based on water quality criteria in the receiving waters themselves. Further investigations regarding input data, complementary measurements, and an evaluation of the validity for substances other than nutrients are recommended to improve the accuracy of calculations.

**Keywords:** stormwater quality and treatment; StormTac Web; OECD; receiving water quality criteria; Water Framework Directive; required treatment



## 1. Introduction

The EU Water Framework Directive (WFD), adopted in 2000 [1], was promulgated to protect surface waters in Europe. The directive commits European Union member states to achieve good status in all surface waters (rivers, lakes, transitional water, coastal water, and sea). To meet the environmental quality standards (EQS), i.e., the goals of good status in surface waters, urban stormwater management must be designed with considerations given to the required pollutant treatment removals needed to protect the receiving waters according to the WFD.

The WFD recommends setting up EQS preferably by including both acute and chronic effects. One goal is to limit the discharge of specific substances to certain maximum concentrations or total loads [2].

Poikane et al. [3] reviewed the nutrient criteria used in Europe under the WFD. Recently, the focus has shifted from the assessment of ecological status towards identifying management measures to achieve good status. Selecting appropriate nutrient criteria is important to enable management measures needed to achieve good ecological status. For lakes, all 26 EU member countries use total phosphorus (TP) criteria, and only 16 countries use total nitrogen (TN) criteria. For rivers, 24 countries use TP criteria, and 17 countries

employ annual mean and median TP concentrations. Across Europe, good–moderate lake TP thresholds are described by a median TP concentration of 27.5 µg/L, varying from 5 to 500 µg/L. The corresponding values for rivers are 100 (8–660) µg/L. The median value for rivers using regression models is 45 µg/L [3].

In Europe, there are no specific guidelines that regulate stormwater effluents from separate stormwater sewer systems to achieve a good status in the receiving waters. Furthermore, the focus is on nutrients and solid particles, but other pollutants also need attention [2]. In both Denmark and Sweden, different strategies are used by different municipalities, which often hire consultants to undertake the required calculations.

A review of 37 Danish discharge permits was performed in 2014–2018. Some discharge permits referred to expected standard removal efficiencies for the chosen control techniques expressed relative to the inlet concentrations, with expected yearly removal loads or concentration levels below a specific value. Many of the reviewed permits included no quantitative or operational requirements concerning water quality. According to a ruling made by the Court of Appeal in 2018 (Court of Appeal under the Danish Ministry of Environment and Food, 2018), general statements are not sufficient; operational requirements should be made to ensure that the general intent of the WFD is carried out [2]. The environmental objectives were only referenced for ecological status in the permits, whereas the chemical status was rarely mentioned. An evaluation of catchment runoff load was performed in some permits, using typical values for average annual concentrations, mainly for the most frequently measured concentrations of TN, TP, and COD in the receiving waters. These were then compared to the estimated removal efficiency (%) of the chosen type of treatment facility to justify that the effluent quality would not hinder attaining the receiving waters objectives.

Actual substance-specific effluent limit values were used in only two of the reviewed discharge permits, neither of which considered short-term or long-term effects. Furthermore, no tools were presented with respect to how to work with such discharges at a strategic level. The permits were largely focused on water quantity issues [4]. For those permits that did set up additional requirements for water quality, there was no proven methodology with relevant uncertainty levels used to evaluate the calculated pollutant load exported from the catchment, to evaluate the treatment facility performance and evaluate whether the environmental objectives in the receiving waters were met. The review pointed out the importance of effect-oriented assessments to account for the total pollutant load to the receiving waters, as well as the need to perform point-by-point estimates rather than designing the treatment measure only from data at a single discharge point [2].

Many models are used outside of Europe to simulate runoff pollutant loads to and/or quality in receiving waters, such as QUAL2E, SENTWA, SIMCAT, and the Vollenweider OECD model [5]. Other examples found in the literature review include the SWAT [6] and HYPE [7] models.

However, among all of these models, only the mentioned Vollenweider OECD model was found to be used specifically for the calculation of required annual treatment load (kg/year) to achieve set critical (limit) annual concentration criteria in the receiving water [4]. The other model descriptions do not include "required pollutant loads", "required treatment", and "acceptable loads" [5–7] and therefore fall outside of the specific type of model or method we searched for and describe in this study.

According to the European Commission [5] there are no other available tools for relating stormwater effluent data to the ambient water quality guidelines within the EU Water Directive, i.e., "the original points of the directive, assessing the pressures that cause impact on biology and ecological status, are not covered by the available tools". In most cases, stormwater effluent data are directly compared to criteria concentrations in the receiving water body; whereas such an approach can guide the design of remedial works, it cannot provide a quantified required treatment removal.

Another method identified in the literature for calculating acceptable load is the TMDL method (total maximal daily loads), with the objective "to determine the loading capacity of the waterbody and to allocate that load among different pollutant sources so that the

appropriate control actions can be taken, and water quality standards achieved". "The TMDL process is important for improving water quality because it serves as a link in the chain between water quality standards and implementation of control actions designed to attain those standards" [8]. This method is used to calculate daily pollutant loads, which can be useful for non-conservative pollutants (e.g., fecal bacteria and COD) causing acute effects in the receiving waters. The method can be integrated for the whole year and used for conservative pollutants (e.g., phosphorus, metals, and solids) causing chronic effects.

In a literature search, we found four available methods that have recently been employed for calculation of required treatment of pollutants (kg/year) discharged to receiving waters, i.e., tools used to relate stormwater effluent data to the ambient water quality guidelines within the EU Water Directive:

Method 1. The StormTac Web model [9,10], which includes a receiving waters model. If there no measured water quality data are available, the model uses the Vollenweider OECD concentration model [11];

Method 2. The Vollenweider OECD load model [11];

Method 3. The model developed by the Swedish Water Authorities and the Swedish Meteorological and Hydrological Institute (SMHI) [12,13]; and

Method 4. Effluent concentration criteria [14].

These methods are described later in the Methods section. For some of the methods, a derivation of new equations of the methodology is suggested to enable calculation of annual acceptable load (kg/year) and annual required treatment load (kg/year) for different types of receiving water and substances, i.e., not only nutrients but also priority substances listed in the Water Framework Directive [1]. It is also necessary to compile and derive different equations depending on whether there are available measured concentration data in the receiving waters (if the concentrations need to be calculated).

The first two presented methods use the Vollenweider model [11] as a basis for the calculations; the novelty lies in the fact that in the first method, the empirical coefficients have been adjusted and compiled for substances other than nutrients [14]. These calculations can be performed for both cases with or without measured concentrations in the receiving water body [10] and with different coefficients for each substance and type of receiving water, such as watercourses and lakes.

The third method was developed by and is used by the Swedish Authorities and SMHI; the chemistry part of the method is based on the same type of equations as the other two methods. The load part of the method can be used without chemistry data. The fourth method can be used if no receiving water calculations are carried out based on concentrations in the receiving water; we derived a new method for calculating the required treatment capacity through the use of effluent criteria.

All methods except Method 3 were developed for use in countries worldwide, but the focus in the literature study was to identify methods to be used within the EU WFD. The compiled Methods 1–3, quantifying the need for treatment, have in common that they are based on the status classification according to the prevailing environmental quality standards. The models use the difference or ratio between the measured average phosphorus concentration and the guideline value, calculated according to the Swedish Marine and Water Authority guidelines [15], in which the calculated reference value (background concentration) is divided by the class boundary between moderate and good status. Both Methods 1 and 3 use typical stormwater concentrations from the StormTac Database [16].

In this study, we provide a general description of the compiled methods, as well as the differences and similarities between the methodologies in terms of calculating receiving water needs for treatment according to the Water Framework Directive [1]. Furthermore, in this study, we aimed to present and test the four selected methods in a case study focusing on total phosphorus as a model substance and review their use, advantages, and limitations.

The present study is relevant and timely because we have noted considerable differences in methodologies used in action plans, substantially affecting the size of treatment

needs and therefore the costs of control measures. It is also relevant to present possible operational methods that may be of wider use within the WFD for quantification of the requirements to meet the objectives of the Directive.

The proposed methods relate to the "Sustainable Development Goals, SDGs" of the United Nations [17], particularly "SDG6-Ensure availability and sustainable management of water and sanitation for all". The aim of the methods described herein is to calculate the required load reduction to protect receiving waters from pollutants and to ensure a good ecological status. If the required actions are performed, they will ensure sustainable management of water.

## 2. Methods

To compile different methods used in urban stormwater management considering the required treatment need for the receiving waters according to the Water Frame Directive, a search of the international literature was performed. A dialogue was also conducted with Swedish stakeholders, such as the Agency for Marine and Water Management, the Country Administrative Board, and municipalities (e.g., Stockholm, Järfälla, and Gothenburg), to increase the understanding of the methods applied to different projects. The methods were applied in a case study to highlight differences, similarities, advantages, and disadvantages of individual methods.

The main equations used in the tested methods are included, together with key input data and parameters for model calibration/validation.

### 2.1. Method 1: StormTac Web's Receiving Waters Model, Including the Vollenweider OECD Concentration Submodel

Key input data: corrected annual precipitation, area and volumetric runoff coefficients per land use, typical concentrations of stormwater and baseflow per land use, area of the receiving water, measured concentrations in the receiving water (if available), design data from existing stormwater treatment facilities (if available), and point loads.

Parameters for model calibration/validation: measured concentrations in the receiving water (if available).

In StormTac Web, the stormwater runoff is calculated as a product of the annual precipitation, the catchment area, and the volumetric runoff coefficients ($\varphi$vol) defined for specific land uses. Pollutant loads in stormwater are estimated by multiplying the annual runoff flow rate by the typical pollutant concentrations corresponding to the catchment land uses [9]. Hence, the catchment is divided into subcatchments with homogenous land use. Concentrations of pollutants in stormwater and baseflow from such land uses can be supplied by the user from locally available data or adopted as typical (default) data from the StormTac Database [16]. The typical concentrations are based on continually updated flow-weighted stormwater samples, which generally produce significantly higher loads than those based on grab sampling. The baseflow is included to calculate the load on the receiving waters, i.e., dry weather runoff, infiltrating groundwater and connected drainage water, and outflowing groundwater. Loads from baseflow can account for a relatively large portion of the total load, depending on the substance and how much of the catchment area is classified as urban land use.

Two papers presented the tool, addressing the issues of calculating pollutant loads, designing stormwater treatment facilities, evaluating their performance, and estimating uncertainties in pollutant calculations [16,18].

The receiving water model in StormTac Web is a tool used to estimate the pollutant concentration/load in/to receiving water; this model output is affected by land uses, possible point loads, and treatment facilities located within the catchment area. To evaluate a receiving water pollutant load, receiving water limit concentrations (i.e., critical concentrations in the receiving waters) are used, e.g., in the form of environmental quality standards (EQS). The model then calculates the mass of annual pollution (kg/year) that may be discharged into the receiving waters without exceeding the annual limit concentrations—in other

words, the so-called acceptable load (kg/year). Based on the receiving waters acceptable load (kg/year) and the total pollution load (kg/year) discharged into the receiving waters, a treatment requirement (kg/year) is calculated. The treatment requirement (kg/year) indicates how much the pollution load (kg/year) must be reduced to achieve an acceptable load (kg/year) to the receiving waters.

The receiving water model shows, among other things, what annual pollutant concentration and/or load of pollutants (kg/year) may be transported to the receiving waters without exceeding the limit concentrations.

The general applied method for calculating the loads in StormTac Web was scientifically reviewed as part of a PhD thesis [9]. The receiving waters model complemented the StormTac Web model and has been presented at conferences and in associated conference papers [10,14,19]. Since the method was developed, it has been continuously updated, revised, and calibrated against new measurements for a large number of watercourses and lakes, as well as employed in a large number of projects.

The required treatment is based on the calculated total load to the receiving waters, measured concentrations in such waters, and the limit value (see Equations (1)–(4)). The required treatment is calculated by Equation (1).

$$\Delta L = L_{in} - L_{acc} \tag{1}$$

where:

$\Delta L$ = required treatment load, by which the pollutant load needs to be reduced to reach the acceptable load $L_{acc}$ (kg/year);
$L_{in}$ = total load (kg/year); and
$L_{acc}$ = acceptable load (kg/year).

The receiving waters model in StormTac Web uses the calculated total load to the receiving waters by means of Equation (2). This method is referred to as Method 1a in the case study.

$$L_{in} = L + L_b + L_a + L_{point} \tag{2}$$

where:

$L$ = load from stormwater (kg/year);
$L_b$ = load from baseflow, dry weather flow runoff, in-leaking groundwater, and connected drainage water (kg/year);
$L_{point}$ = point flow within the catchment area (reduced load in facilities, wastewater treatment plants, upstream receiving waters, etc.); reduced loads in facilities are inputs with negative signs in the equation, as they reduce the total load on the receiving waters (kg/year); and
$L_a$ = atmospheric deposition directly on the water surface (kg/year).

To calibrate the calculated total load ($L_{in}$), the measured concentration and calculated outflow ($L_{out}$) are used (see Equations (3)–(5)). This method is referred to as Method 1b in the case study.

$$L_{in} = L + L_b + L_a + L_{point} - L_{ret} \tag{3}$$

$$L_{ret} = L_{in} - L_{out} \tag{4}$$

$$L_{out} = \frac{Q_{out} \cdot C_{rec}{}^{m}}{1,000,000} \tag{5}$$

where:

$L_{out}$ = outflow pollutant load from the receiving waters (kg/year);
$Q_{out}$ = outflow from the receiving waters (m$^3$/year);
$C_{rec}{}^{m}$ = measured pollutant concentration in the receiving waters (µg/L); and
$L_{ret}$ = load retention (kg/year) representing a net yearly load entering the sediments or net internal pollutant load transferring from the sediments to the water body. This term refers

to pollutant retention in the catchment area (ditches, watercourses, etc.) and retention in the receiving waters themselves.

The load retention ($L_{ret}$) is the difference between the total load to the receiving waters ($L_{in}$) and the load leaving from the receiving waters ($L_{out}$) and comprises the retention in the catchment area and the receiving waters, inaccuracies regarding the input describing the catchment area, i.e., the area per land use, etc., and over- or underestimation of land-use-specific typical concentrations and volumetric runoff coefficients. The load retention is considered to be an important part of the receiving waters model, which can, to a large extent, affect the required treatment to achieve an acceptable load. Significant annual retention of phosphorus and other substances in watercourses and lakes can be caused by TP entry into sediments [20]. The use of mass balance to calculate retention is a very useful method for calculating the overall function of the receiving waters [21]. Each catchment area has a unique mass balance, owing to many site-specific parameters that affect load and retention, such as vegetation, soil properties, precipitation, land use, and the size of the catchment area [22].

The acceptable load ($L_{acc}$) refers to how much load ($L_{in}$) the receiving waters can receive without the measured concentration in the receiving waters exceeding the critical concentration ($C_{cr}$). The acceptable load to the receiving waters is calculated by Equation (6) when measured concentration data are available. This method is referred to as Method 1b in the case study.

$$L_{acc} = \frac{C_{cr} \cdot L_{in}}{C_{rec}{}^m} \tag{6}$$

where:

$C_{cr}$ = critical annual pollutant concentration in the receiving waters for (avoiding) negative effects (μg/L).

The acceptable load to the receiving waters is calculated by Equation (7) when measured concentration data are not available, as presented in [14]. This method is referred to as Method 1a in the case study.

$$L_{acc} = \frac{C_{cr} \cdot L_{in}}{C_{rec}} \tag{7}$$

where:

$C_{rec}$ = calculated pollutant concentration in the receiving waters (μg/L).

The calculated receiving waters concentration ($C_{rec}$) in Equation (7) is derived from the Vollenweider OECD concentration model [11] presented in [14] (see Equation (8)). This method is referred to as Method 1a in the case study.

$$C_{rec} = 1000 \cdot x_j \cdot \left( \frac{1000 \cdot C_{in}}{(1 + t_{dr}{}^{0.5})} \right)^{yj} = 1000 \cdot x_j \cdot \left( \frac{1000 \cdot L_{in}}{Q_{in} \cdot (1 + t_{dr}{}^{0.5})} \right)^{yj} \tag{8}$$

where:

$x_j$, $y_j$ = empirical coefficients for pollutant $j$, with different coefficients for lakes and watercourses;

$C_{in}$ = mean pollutant concentration in the inflows to the receiving waters (μg/L);

$Q_{in}$ = total inflow to the receiving waters (m$^3$/year); and

$t_{dr}$ = receiving waters residence time ($t_{dr} = V_{rec}/Q_{out}$ (year)).

Based on the application of Equation (8) in Swedish case studies, we have calculated mean values of $x_j$ and $y_j$ for different pollutants for 17 Swedish watercourses, where for phosphorus (TP), empirical coefficients $x_P = 0.72$ and $y_P = 1.04$ were established and used in the case study and implemented in StormTac Web v22.3.2 (to be further updated when more data have been processed).

The OECD model was designed for lakes, not for watercourses. However, we calculated specific coefficients for case studies consisting of watercourses, thereby taking into

consideration their shorter residence times ($t_{dr}$), as there are specific OECD coefficients to be used for shallow lakes and reservoirs that also have shorter residence times [11,23].

The corresponding values for 6 calculated Swedish lake case studies implemented in StormTac Web v22.3.2 were $x_P = 0.74$ and $y_P = 0.86$. For comparison, Vollenweider and Kerekes [11] presented the corresponding coefficients for combined OECD data as $x_P = 1.55$ and $y_P = 0.82$ ($n = 87$), from Nordic data as $x_P = 1.12$ and $y_P = 0.92$ ($n = 14$), and from shallow lakes and reservoirs as $x_P = 1.02$ and $y_P = 0.88$ ($n = 24$).

### 2.2. Method 2. The Vollenweider OECD Load Model

Key input data: calculated or measured annual outflow from the receiving water and/or the residence time, measured mean water depth and area of the receiving water, and measured concentrations in the receiving water (if available).

Parameters for model calibration/validation: measured concentrations in the receiving water (if available).

The Vollenweider OECD model is a widely used and accepted approach for modelling nutrients and is one of several proposed screening tools that can be used within the EU Water Framework Directive [5]. It was designed for lakes, including shallow lakes and reservoirs with shorter residence times [11]. We assumed that it can also be used for watercourses with shorter residence times, as applicable to our case study. It was previously used for reservoirs by Brigault and Ruban [23].

The acceptable load to the receiving waters is calculated in Equation (9), derived from the OECD load model [11]. The right side of the equation was derived from $A_{rec} = V_{rec}/h$ and $t_{dr} = V_{rec}/Q_{out}$ and was used in the case study, as the input data of mean water depth and the area of the watercourse are uncertain. This method is referred to as Methods 2a and 2b in the case study.

$$L_{acc} = \frac{C_{cr} \cdot h \cdot A_{rec}}{10 \cdot t_{dr}} = \frac{C_{cr} \cdot Q_{out}}{1,000,000} \tag{9}$$

where:

$h$ = mean water depth in the receiving waters (m); and
$A_{rec}$ = area of the receiving waters (ha).

The calculated treatment need is based on calculated or measured concentration in the receiving water, the calculated outflow, and the limit concentration value.

If there is no measured receiving waters concentration, Equation (10) can be used, and $C_{rec}$ can be calculated from Equation (8). The equation can be derived from Equation (1) when $L_{in} = L_{out} = Q_{out} \cdot C_{rec}$, as $L_{in}$ in Equation (3) includes $L_{ret}$ and $L_{in} = L_{out}$. This method is referred to as Method 2a in the case study.

$$\Delta L = \frac{Q_{out} \cdot (C_{rec} - C_{cr})}{1,000,000} \tag{10}$$

The treatment need can be calculated according to Equation (11) using the measured receiving waters concentration. This method is referred to as Method 2b in the case study.

$$\Delta L = \frac{Q_{out} \cdot (C_{rec}^{m} - C_{cr})}{1,000,000} \tag{11}$$

### 2.3. Method 3: The Model Developed by the Swedish Water Authorities and SMHI

Key input data: corrected annual precipitation, area and volumetric runoff coefficients per land use, typical concentrations of stormwater and baseflow per land use, and measured concentrations in the receiving water (if available).

Parameters for model calibration/validation: measured concentrations in the receiving water (if available).

To calculate the required treatment need according to the WFD, the model developed by the Swedish Water Authorities and SMHI (the Swedish Meteorological and Hydrological Institute) takes two approaches using either the water chemistry or the calculated pollutant load. Good ecological status regarding the nutrient total phosphorus (TP) is determined according to the method defined by an ecological quotient (*EQ*) [12]. If $EQ \geq 0.5$, the receiving waters achieved good status (see Equations (12) and (13)).

$$EQ_{load} = \frac{L_{back}}{L_{ext}} \geq 0.5 \tag{12}$$

where:

$EQ_{load}$ = ecological quotient based on load;
$L_{back}$ = background load, defined as load with no human impact (kg/year); and
$L_{ext}$ = external pollutant load on the receiving waters (kg/year).

$$EQ = \frac{Crec^m}{C_{cr}} \geq 0.5 \tag{13}$$

where:

$C_{rec}{}^m$ = measured pollutant concentration in the receiving water (µg/L).

The water chemistry is used when such data are available, and the measured pollutant concentration can be explained by the calculated load, i.e., if *EQ* is not significantly smaller than $EQ_{load}$. The load approach may be used if $EQ_{load}$ is significantly smaller than *EQ* or if $C_{cr}$ is considered uncertain [12].

The pollutant load used in both approaches is calculated with the Pollutant Load Compilation model (PLC 6.5) based on annual precipitation, catchment area, and typical values per land use of volumetric coefficients and pollutant concentrations. PLC 6.5 calculates the nutrient load for phosphorus and nitrogen. A detailed calculation is performed regarding the load from natural land, such as agricultural land, and various crops, soil types, fertilization, and leakage between seasons are considered [24]. For urban areas, only one land use is employed: "urban". Retention of nutrients is included in the calculations, defined as the difference between the total load to the receiving waters and the load leaving the receiving waters [25].

2.3.1. Required Treatment Based on the Water Chemistry

By assuming the same relative reduction for the external loads as for the concentration in the receiving waters, an acceptable load ($L_{acc}$) is calculated according to Equation (14), rearranged from [12].

$$L_{acc} = \frac{C_{cr} \cdot L_{ext}}{C_{rec}{}^m} \tag{14}$$

where:

$L_{ext}$ = external pollutant load on the receiving waters (kg/year).

The required treatment is calculated from Equation (15).

$$\Delta L = L_{ext} - L_{acc} \tag{15}$$

This need for improvement, expressed as $\Delta L$, represents the reduction in the external load that is needed to achieve good status regarding nutrients according to Equation (10). This method was not used in the case study.

2.3.2. Required Treatment Based on the Pollutant Load

The external load ($L_{ext}$) to the receiving waters is divided into two components: the anthropogenic load ($L_{ant}$) and the background load ($L_{back}$). The background load is based on a hypothetical scenario to quantify the leakage without human impacts. The following adjustments are made [24]: (1) point sources, such as small drains and stormwater loads,

are set to zero; (2) urban areas and paved surfaces are replaced by open land; (3) forest clearings are replaced by forest; (4) the internal load of phosphorus is removed; and (5) for agricultural land, cultivated crops are replaced by a background value for agricultural land calculated by SLU. This method was used in the case study.

The acceptable load is defined according to Equations (16) and (17), rewritten from Lampa [12]:

$$EQ_{Load} = \frac{L_{back}}{L_{ext}} \geq 0.5 \rightarrow L_{ext} \leq 2 \cdot L_{back} \tag{16}$$

where:

$L_{back}$ = background load (kg/year).

Equation (16) can be written as Equation (17), assuming $EQ_{load}$ = 0.5.

$$L_{acc} = 2 \cdot L_{back} \tag{17}$$

The required treatment is defined according to Equation (18).

$$\Delta L = L_{ext} - 2 \cdot L_{back} \tag{18}$$

This need for improvement, expressed as $\Delta L$, represents the reduction in the external load that is needed to achieve good status regarding nutrients according to Equation (10).

The required load reduction can be reduced if the anthropogenic load ($L_{ant}$) is smaller than the calculated required treatment according to Equations (15) and (18).

### 2.4. Method 4: Outlet Concentration Criteria

Key input data: corrected annual precipitation, area and volumetric runoff coefficients per land use, typical concentrations of stormwater and baseflow per land use, and critical pollutant stormwater concentration in the discharge to the receiving waters

Parameters for model calibration/validation: measured outlet concentrations to the receiving water (if available).

This method uses guidelines as concentrations in a mix of stormwater and baseflow effluents, reaching the receiving water. These outlet concentration criteria can be used as a screening tool for action planning and for identification of action needs, but the total watershed loads (kg/year) on the receiving waters also need to be calculated to quantify the required treatment load ($\Delta L$) for the receiving waters. In general, load-based planning is more relevant for assessment of the impact on receiving waters than just studying concentrations at the point of discharge to the receiving waters, which does not include the receiving waters criteria themselves. The outlet concentrations refer to an annual average concentration, not a maximum concentration, because the threshold values for stormwater refer to annual mean concentration values. A relatively high outlet concentration ($\mu$g/L) can yield a small load (kg/year) on the receiving waters if the catchment area is small and thus produce a relatively small concentration change in the receiving waters compared to other areas with lower concentrations but larger catchment areas. For the receiving waters, it is most important to reduce the total external load so that the receiving waters concentration is reduced sufficiently, taking into account the water quality goal of the WFD.

It is important to distinguish between concentrations in the stormwater discharge points to the receiving waters and concentrations in the receiving waters. Concentrations in stormwater and guidelines for stormwater discharges in the form of limit outlet concentrations should not be directly compared with guidelines for receiving waters in the form of limit concentrations in the receiving water body. In many cases, the guideline values in the receiving waters are also lower than the concentrations of the incoming water; this also applies to certain substances from natural lands. In the receiving waters, treatment processes and dilution of polluted influents take place. Threshold values for stormwater can therefore be set higher than the guidance values for the receiving waters.

However, in some individual cases, it may be difficult to estimate only the need for action based on pollutant loads, e.g., if the studied load is very small compared to the total load on very large receiving waters. Then, the load from the area does not have any major significance for the receiving waters, independent of the implemented measures. Nevertheless, measures may be relevant because in the long term, it is advantageous to remedy many parts of the catchment area's load so that measurable effects in the receiving waters are ultimately achieved. In such cases, it is difficult to estimate the load that needs to be reduced. An option in that case is to follow the proposed guideline values for stormwater discharges in the form of concentrations.

The outlet concentration criteria can be used for a subcatchment to calculate the required treatment (kg/year) or for the whole watershed area to calculate the total required treatment for the receiving waters.

First, the total annual mean concentration (stormwater + baseflow) in the outlet discharge is calculated or measured with flow proportional sampling. Second, threshold concentration values for stormwater (including baseflow) discharge are used, taking into account the type of area and the receiving waters. If the outlet concentration criteria are exceeded for any relevant substance, the required treatment for the substance is calculated. Measures are designed to achieve the required treatment effect; thus, the concentrations at the point of discharge are below the outlet concentration criteria.

This method employing calculated or measured effluent concentrations and effluent criteria concentrations is used in some countries and by some municipalities, e.g., in Sweden. Other municipalities use the load-based criteria (Methods 1–3).

Equation (19) can be used to calculate the required treatment load based on the difference between the inlet concentration and effluent concentration criteria.

$$\Delta L = \frac{Q_{in} \cdot (C_{in} - C_{cr,sw})}{1,000,000} \tag{19}$$

where:

$C_{cr,sw}$ = critical pollutant stormwater concentration in the discharge to the receiving waters (μg/L).

In order to calculate not only the subcatchment required treatment load but also the total load (and concentration) required for the entire receiving waters, the total load from the whole watershed area needs to be calculated, as performed for our case study.

### 3. Case Study Catchment

The Bällstaån watercourse and its catchment were selected for a case study for the following reasons: (i) it represents a typical Swedish urban catchment with respect to a variety of land uses; (ii) it features both existing and planned stormwater treatment facilities; (iii) stream water quality data are available. In this section, we present the required input data for the methods described above.

The nutrient total phosphorus (TP) was selected for the pollutant calculations for the following reasons. There are no total nitrogen (TN) thresholds used in Sweden; nutrients are the most commonly measured water quality parameters in Swedish lakes, rivers, and other watercourses, and two of the four described methods focus on nutrients (Methods 2–3), whereas only one of the methods surveyed was previously applied to TP in Sweden (Method 3).

Bällstaån is a 10 km long [13] natural watercourse located in Järfälla and Stockholm municipalities in Sweden, with a total catchment of 3600 ha (Figure 1). The catchment area mostly consists of various types of residential and industrial areas, as well as the downtown. Bällstaån is included in a regional environmental monitoring program established by the Swedish County Board and includes water chemistry sampling at the outlet once a month. The water quality of Bällstaån varies considerably, with high concentrations of phosphorus and nitrogen [26]. Figure 1 shows the existing stormwater treatment facilities and areas with combined sewers within the watershed.

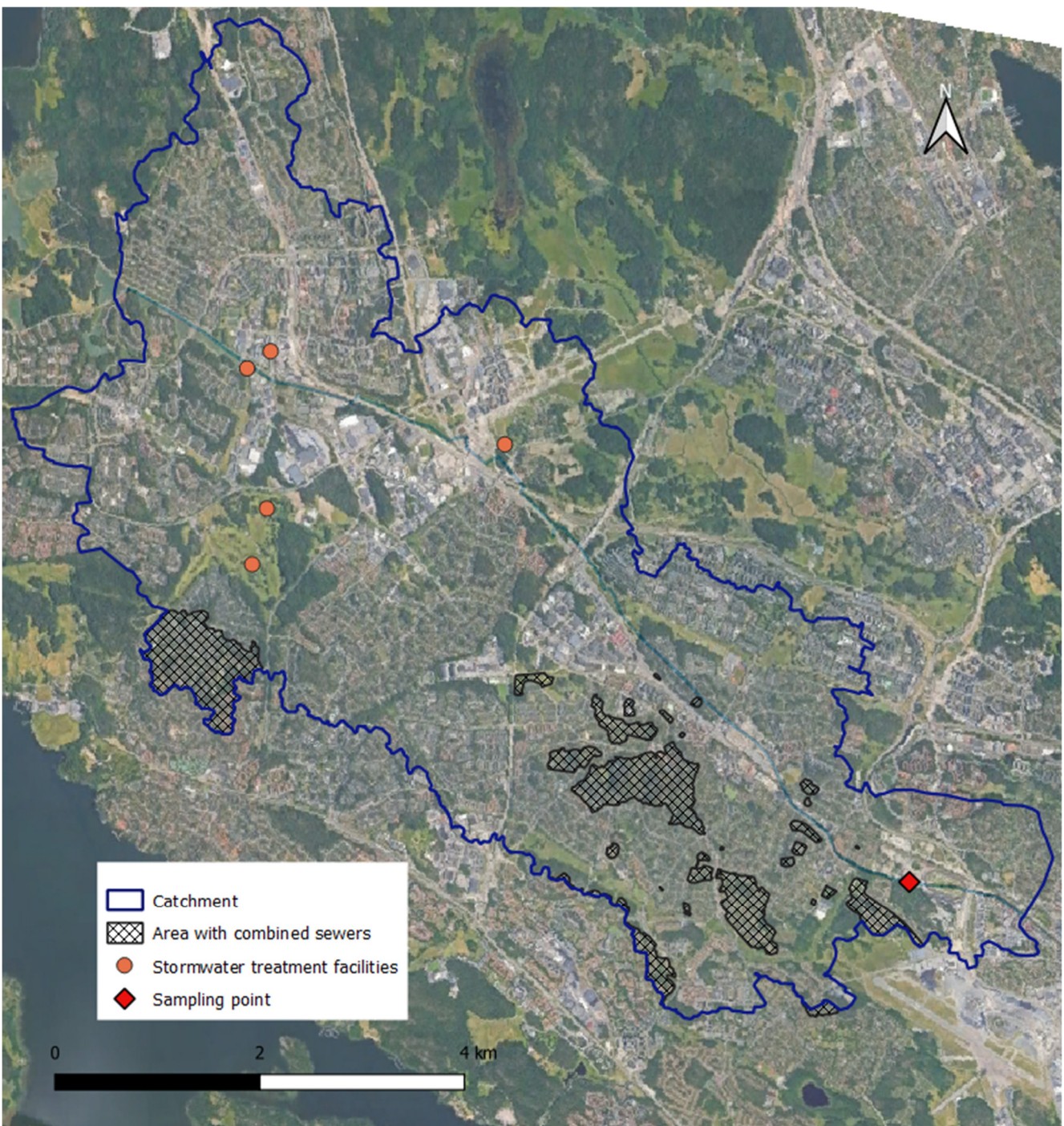

**Figure 1.** Map of the total catchment of Bällstaån. Legend: the total catchments boundaries of Bällstaån are colored blue, and the parts of the catchment representing combined sewers are colored grey. Orange dots represents the existing stormwater treatment facilities, and the red square marks the sampling point. Orthophoto: Image Landsat Copernicus, Google Earth Pro.

The catchment area is 3600 ha and was discretized into 42 subcatchments representing various land uses (Table 1).

**Table 1.** Area per land use within the catchment. Typical volumetric runoff coefficients were used in runoff calculations. ADT = average daily traffic (vehicles/day).

| | Subcatchment Land Use (-) | Runoff Contributing Area (ha) | Combined Sewer Area (ha) | Volumetric Runoff Coefficient (-) |
|---|---|---|---|---|
| 1 | Agricultural property | 31.3 | 1.0 | 0.26 |
| 2 | Allotment area | 3.0 | | 0.15 |
| 3 | Asphalt surface | 3.4 | | 0.80 |
| 4 | Cemetery | 7.7 | | 0.10 |
| 5 | Downtown area | 65.2 | | 0.60 |
| 6 | Forest | 560 | 13 | 0.15 |
| 7 | Gas station | 4.6 | | 0.80 |
| 8 | Golf course | 66.1 | | 0.10 |
| 9 | Gravel cover | 6.7 | | 0.40 |
| 10 | Hospital | 3.4 | | 0.70 |
| 11 | Industrial area | 300 | 1.6 | 0.50 |
| 12 | Larger parking area | 4.3 | | 0.80 |
| 13 | Loading dock | 2.1 | | 0.80 |
| 14 | Meadow | 12 | | 0.10 |
| 15 | Metal recycling center | 3.4 | | 0.80 |
| 16 | Mixed green area | 540 | 25.6 | 0.12 |
| 17 | Mixed multifamily area and downtown, suburbs | 4.0 | | 0.60 |
| 18 | Mixed residential area | 580 | 180 | 0.29 |
| 19 | Multifamily area | 380 | 2.8 | 0.40 |
| 20 | Office area | 51.3 | | 0.50 |
| 21 | Park grounds | 55.4 | | 0.10 |
| 22 | Parking | 12.1 | | 0.80 |
| 23 | Railway embankment | 19.8 | | 0.50 |
| 24 | Recreational area | 37.2 | | 0.25 |
| 25 | Recycling center | 1.6 | | 0.70 |
| 26 | Residential area | 410 | | 0.25 |
| 27 | Road 1 (ADT 500) | 0.16 | | 0.80 |
| 28 | Road 2 (ADT 1500) | 8.2 | | 0.80 |
| 29 | Road 3 (ADT 3500) | 21.2 | | 0.80 |
| 30 | Road 4 (ADT 7500) | 19.3 | | 0.80 |
| 31 | Road 5 (ADT 12,500) | 28.8 | | 0.80 |
| 32 | Road 6 (ADT 20,000) | 8.9 | | 0.80 |
| 33 | Road 7 (ADT 15,000–30,000) | 7.2 | | 0.80 |
| 34 | Road 8 (ADT 25,000) | 7.1 | | 0.80 |
| 35 | Road 9 (ADT 30,000) | 5.8 | | 0.80 |

**Table 1.** *Cont.*

|  | Subcatchment Land Use (-) | Runoff Contributing Area (ha) | Combined Sewer Area (ha) | Volumetric Runoff Coefficient (-) |
|---|---|---|---|---|
| 36 | School campus | 94.7 | 2.6 | 0.45 |
| 37 | Surface waters | 5.0 |  | 1.0 |
| 38 | Terraced house area | 210 |  | 0.32 |
| 39 | Thermal power station with storage areas and trafficked surfaces | 1.2 |  | 0.70 |
| 40 | Vacation cottage development | 8.0 | 0.54 | 0.15 |
| 41 | Waste dump site | 0.024 |  | 0.80 |
| 42 | Wetlands | 10 |  | 0.20 |
|  | Total area (ha) | 3600 | 230 |  |

For watercourses and phosphorus (P) in Sweden, it is recommended to calculate a mean value for a 3-year period; in cases in which few data are available or the variation between years is large, 6 years of data may be used [15]. Mean values for a 3-year period were calculated from January 2019 to December 2021 (Table 2) [26].

**Table 2.** Stream water quality in Bällstaån presented as means from January 2019 to December 2021.

| TP (µg/L) | Ca (µg/L) | Mg (µg/L) | Cl (µg/L) | AbsF * (µg/L) | Elevation ** (m.a.s.l.) |
|---|---|---|---|---|---|
| 99 | 63,000 | 7200 | 78,000 | 0.055 | 3.0 |

* Filtered absorbance, ** elevation of the sampling point.

Swedish limit values are assumed as thresholds for the nutrient criteria pertinent to "good ecological status". The input data compiled in Table 2 are needed to calculate a critical phosphorus concentration in the receiving waters [15]. However, other pollutants, such as Cu, Zn, Cd, Cr, and Ni, can also be calculated with Methods 1 and 4, for which the critical concentrations are given in the WFD, not requiring other measured data or calculations.

The TP limit concentration value ($C_{cr}$) was calculated as 47 µg/L based on the data from Bällstaån presented in Table 2 and justified according to a factor ("*Pjo*") representing the original share of agricultural land (farmland) according to the Swedish methodology [15].

Precipitation input data were obtained from SMHI Station Observatorielunden Stockholm and corrected to yield an annual value of 601 mm/year for the 30-year normal during the period of 1991–2020 [27].

The annual outflow from Bällstaån was calculated with StormTac Web to be $9.5 \cdot 10^6$ m$^3$/year, representing a mean value for the period of 1991–2020 [27] used in the calculations for the case study with various methods to yield comparable calculated values of the acceptable and required loads. Calculation of the annual flow included stormwater runoff and baseflow. The methodology for the calculation has been scientifically reviewed (e.g., [9]) and showed satisfactory agreement with annual flows calculated by other models for the same case study [28] and other case studies and flow datasets.

To calculate the total load to the receiving waters according to Method 1, the StormTac Web receiving waters model was used, and in Method 3, the model developed by the Swedish Water Authorities and SMHI was applied. For these calculations, the following types of land-use-specific data were needed: volumetric runoff coefficients and typical water quality parameter concentrations for stormwater and baseflow. For the calculated results produced by Method 1, these data were adopted from the StormTac Database and accessed in October 2021.

## 4. Results and Discussion

### 4.1. All Methods (1–4)

Table 3 presents the calculated load, acceptable load, and required treatment of TP for the case study and the presented methods.

**Table 3.** Calculated total loads, acceptable loads, and required treatment capacity (kg/year) of TP for Bällstaån and various calculation methods.

| | Receiving Water Concentration (μg/L) | Total Load (kg/year) | Acceptable Load (kg/year) | Required Treatment (kg/year) |
|---|---|---|---|---|
| Method 1a. StormTac Web receiving water model, including the Vollenweider OECD concentration model. Uncalibrated using calc. $C_{rec}$. | 94 | 1400 | 700 | 700 |
| Method 1b. StormTac Web receiving water model. Calibrated using meas. $C_{rec}{}^m$ and $L_{ret}$. | 99 | 950 | 450 | 500 |
| Method 2a. Vollenweider OECD load model using calc. $C_{rec}$. | 94 | - | 450 | 450 |
| Method 2b. Vollenweider OECD load model using meas. $C_{rec}{}^m$. | 99 | - | 450 | 490 |
| Method 3. The model developed by the Swedish Water Authorities and SMHI. | - | 730 | 560 | 170 |
| | Calculated effluent concentration (μg/L) | Total load (kg/year) | Effluent concentration criteria (μg/L) | Required treatment (kg/year) |
| Method 4. Effluent concentration criteria. | 150 | 1400 | 80 | 670 |

### 4.2. Methods 1–2

The calculated receiving waters TP concentration of 94 μg/L was close to the measured concentration 99 μg/L. The calculated required TP treatment to achieve good ecological status at 47 μg/L varied between 450 and 700 kg/year, with smaller variations when using a calibrated and measured concentration (490–500 kg/year) than when using uncalibrated and calculated receiving water concentrations (450–700 kg/year).

The total load used in Methods 1a–b presented in Table 3 was reduced by subtracting a TP load of 144 kg/year, calculated with the StormTac Web model, as the load removed in existing stormwater treatment facilities in the Järfälla municipality.

The uncalibrated method (Method 1a) neglects TP retention processes (i.e., assuming $L_{ret} = 0$ in Equation (3)), such as those taking place in treatment facilities, TP retention in natural ponds and ditches, and the net TP retention or internal load to/from the sediments of the watercourse, as well as uncertainties in measured areas representing various land uses within the catchment area and uncertainties in the employed land use specific volumetric runoff coefficients and typical concentrations. Such processes are included after calibration in Method 1b.

### 4.3. Method 3

The required treatment capacity is calculated as 63 kg/year according to the compiled equations based on the load and estimated removal (reduction) in existing stormwater treatment facilities [13]. The much smaller required total treatment of 170 kg/year calculated with Method 3, compared to the corresponding values from the other methods, can be explained by the fact that the employed load-based model for these receiving waters does not include any chemistry data measured in receiving waters, e.g., the measured and critical concentrations. The calculated required treatment value is based on the load

calculations meant to represent both the necessary and the "feasible" treatment values. However, the required treatment would be only approximately 90 kg/year, using the same method as if the stormwater load was reduced with the same estimated removed load of 144 kg/year (used in the StormTac Web model) to account for existing treatment facilities in the Järfälla municipality. The calculated stormwater load was based on Method 1 but using older data from the StormTac Database (2014 version). If the chemistry data had been considered, a larger required treatment load would have been calculated, which also is required for the receiving water concentration to decrease to the limit concentration value. If the same external load was used as in Methods 1–2, the same required load would have been calculated.

### 4.4. Method 4

The outlet concentration criteria are specific to Bällstaån and were set by the Järfälla municipality [29]. The calculated outlet concentration and the required treatment were reduced for a calculated load reduction of 144 kg/year in the existing treatment facilities. The calculations were performed with the StormTac Web model. The method based on outlet concentration criteria does not include data such as the measured receiving waters concentration or the critical concentration but reflects the difference between the calculated effluent concentration and the effluent criteria concentration. The estimation of effluent criteria concentration considerably impacts the required treatment results. In this case, the latter was 670 kg/year and within the same interval as the required treatment calculated with Methods 1–2.

### 4.5. Discussion of the Results Obtained with All the Methods Tested (1–4)

The resulting required treatment values from Methods 1a and 4 are close, mostly as a coincidence explained by the selected effluent criteria in Method 4, which are low in comparison to normally used criteria and in comparison to typical stormwater concentrations from most rural and urban land use catchments [16]. Consequently, such a procedure resulted in a high required treatment capacity for the case study. However, this finding could also be explained by the assumed equal load used in the two methods. Normally, the required load is not calculated when Method 4 is employed in projects, but we presented a suggested method for making this calculation. In most projects that employ Method 4, the required treatment from a studied subcatchment is designed to obtain the calculated outlet concentration from the deigned facility to be equal or smaller than the effluent concentration criteria value, i.e., not considering the total pollutant load on the receiving water and the impact on the receiving water body itself.

Methods 1b, 2a, and 2b achieved results in a similar range, which can be explained by using the same annual outflow, together with the same calculated limit receiving water concentration and a close match between the calculated and measured receiving water concentrations in the case study. When using a larger, uncalibrated pollutant load, the resulting required treatment yielded a larger value, and when employing Method 3b, the concentration data of the receiving water were not used, introducing a risk that the limit concentrations will not be met by designing for small but "feasible" calculated required treatment. The latter could be too small if the total calculated load is too small.

No total load was calculated in the application of Method 2. Load calculations are required to calculate a load from a subwatershed area to the receiving waters and the required load from this area. Such a load can be calculated with Method 1. The advantage of using StormTac Web (Method 1) is that it calculates both the total load and the acceptable load, which can be used to design treatment facilities with the goal of reaching the acceptable load and assessing the reduced load needed to reach the critical pollutant concentrations in the receiving waters.

Retention is calculated in the same way as in Method 3 as in Method 1, as the difference between the total load to the receiving waters and the effluent load from the receiving waters.

The calculations in Method 3 are carried out at a national level, and Sweden's receiving waters consist largely of rural watercourses, for which the method is adapted. The method is not specifically adapted for receiving waters, where stormwater from urban areas accounts for the greatest load, as was also the case in the case study.

The values of calculated required treatment loads for the studied receiving waters are smaller in Methods 1b and 3b than when not accounting for the retention ($L_{ret}$). However, if in another case, the measured concentration ($C_{rec}$*) was lower than the critical concentration ($C_{cr}$), there would still be no treatment need, regardless of the inclusion of $L_{ret}$. If the calculations result in an internal load from the sediment, the inclusion of $L_{ret}$ results in an increased treatment need, in contrast to the presented case study.

All presented methods require relatively little input data. By including a limited number of parameters in calculations of treatment needs, uncertainties are reduced, as each parameter used in a calculation involves a certain amount of uncertainty.

On the other hand, any method that is too simple may not sufficiently reflect the course of events. However, the actual retention within the watershed and receiving waters is included but not calculated by using measured concentrations in the outlet from the receiving waters.

Method 2 has the advantage that it can calculate an estimated treatment need based on only measured concentrations and calculated or measured outflow. It has the disadvantages that it does not consider the calculated load on the receiving waters before and after implementation of control measures and that it does not account for any changes in pollutant retention before and after the implementation of measures. Net sedimentation within the inflow area and in the receiving waters is assumed to be unchanged. However, it is accepted that retention in ditches, natural ponds, and the catchment area, together with retention in the receiving waters, are affected by different site-specific conditions and that it is not a constant mass, but its magnitude is affected by changes in conditions [20,22].

A general assumption is that on an annual basis, retention increases with increased concentrations and loads, with a hypothesis that after control measures are implemented and the load on and concentration are reduced in the receiving waters, pollutant retention decreases. If, instead of this reduced retention, a constant retention is assumed, an underestimation of the need for treatment is obtained. It can be an issue of a strong underestimation of the need for treatment, depending on site-specific conditions in the receiving waters and their catchment area.

We recommend the used of Methods 1–2 to calculate the required treatment of more priority substances than just for nutrients in projects worldwide and the design of required stormwater treatment facilities for the calculated treatment load and based on the receiving water quality rather than on effluent criteria. To obtain calculations, more substances need to be measured in the receiving water, which will also improve the empirical coefficients to be used when there are no quality data available. We suggest the coefficients be used for substances other than nutrients, specific for different types of receiving waters, especially for watercourses and lakes. This should be reviewed and possibly revised by using other types of equations.

We recommend focusing further research on those equations that have been derived and implemented in the StormTac Web model (Method 1). We also recommend inclusion of the equations in models other than StormTac Web to enable linking of the design of treatment to the effects on the receiving waters and to the quality goals of the WFD. More substance data are also needed to improve the calculation of treatment effects of different types of treatment facilities and to calculate the pollutant loads, which is an important basis for the calculation of the required treatment load. The total annual pollutant load and/or the annual flow need to be carefully calculated and should include all loads to the receiving water—not only runoff but also baseflow, groundwater flow, and different types of point loads/flows, as implemented in the StormTac Web model. The methods also need to be adapted for both urban and rural land uses and receiving waters, possibly with more

specific input data for rural catchments and catchments with a large portion of farmland, as the inputs in StormTac Web are more specialized for many types of urban land uses.

## 5. Conclusions

In our literature search, we identified four methods that can be used specifically to calculating annual required treatment loads: (1) the StormTac Web model, including the Vollenweider OECD concentration model; (2) the Vollenweider OECD load model; (3) the model developed by the Swedish Water Authorities and SMHI (the Swedish Meteorological and Hydrological Institute); and (4) the method using effluent concentration criteria.

Several other models were found for calculating pollutant loads and receiving water concentrations, but such models typically cannot calculate a specific annual required treatment load to reach a limit concentration value in the receiving water.

We compiled equations and derived new equations for the four methods that can be used for a specific calculation of the required annual treatment loads.

The required key input data consist of calculated or measured annual runoff and baseflow flows (calculated from the area per land use, volumetric runoff coefficients, and corrected annual precipitation), measured or calculated receiving water concentrations, estimated or calculated criteria concentrations in the receiving water body (Methods 1–2), criteria concentration for effluent from the receiving waters (Method 4), and inputs from existing treatment and other point loads.

The compiled methods were applied for total phosphorus (TP) to the urban watercourse in Bällstaån, Sweden, which requires improvements in water quality to achieve good status. Results showed good agreement among the calculated required treatments, when measured TP was used in Methods 1–2, but elsewhere, there were larger variations. The load approach applied in Method 3 resulted in a substantially smaller required treatment pollutant load. Method 4 neglects the receiving water concentrations and criteria, but the applied calculated concentration and applied effluent criteria resulted in a similar calculated required treatment load.

All four methods can be used for the calculation of the required annual treatment load, but only Methods 1–3 (and only if the chemical part of Method 3 is applied) are based on actual water quality and associated criteria in the receiving waters. The use of the load model in Method 3 and Method 4 can result in an insufficient reduction in loads to and concentrations in the receiving water, especially if the calculated load is underestimated, resulting in the WFD goals of good status not being achieved in the surface waters.

More measurements are recommended of pollutants other than nutrients in the receiving waters to improve the accuracy of the calculations. Methods 1–4, except the load-based part of Method 3, were implemented in the StormTac Web model, which makes it possible to further study the application of the methods in different projects and receiving waters. We recommend using the compiled equations in new or existing operational models in projects to design required treatment in watersheds to receiving waters for those substances with measured concentrations in the waterbody that exceed the criteria concentrations.

This work can be useful and/or beneficial in achieving the UN Sustainable Development Goals, especially SDG 6, as it compiles operative tools for calculating the required treatment load to achieve good status in receiving waters, as exemplified for TP in this article. The tools were also implemented in the StormTac Web-model for other substances of priority within the WFD but can be implemented in other stormwater and/or receiving water models or used as stand-alone equations for the calculation of the total required treatment load and used for other directives or guidelines in other parts of the world.

**Supplementary Materials:** The following are available online at https://www.mdpi.com/article/10.3390/su142215395/s1 Supplementary Material. Original (standard) reference titles that were translated in the References.

**Author Contributions:** Conceptualization, T.L. and A.W.; methodology, T.L. and A.W.; software, T.L. and A.W.; validation, T.L. and A.W.; formal analysis, T.L. and A.W.; investigation, T.L. and A.W.; resources, T.L. and A.W.; data curation, T.L. and A.W.; writing—original draft preparation, T.L. and A.W.; writing—review and editing, T.L. and A.W.; visualization, T.L. and A.W.; supervision, T.L.; project administration, T.L.; funding acquisition, T.L. All authors have read and agreed to the published version of the manuscript.

**Funding:** This work was supported by the DRIZZLE Centre for Stormwater Management, funded by the Swedish Governmental Agency for Innovation Systems (Vinnova) (grant no. 2016-05176).

**Institutional Review Board Statement:** Not applicable.

**Informed Consent Statement:** Not applicable.

**Data Availability Statement:** Not applicable.

**Acknowledgments:** We are very grateful for valuable advice and editing by Jiri Marsalek, formerly National Water Research Institute, Canada, and Luleå University of Technology, Sweden. We also gratefully acknowledge Jonathan Fridesjö and Martin Erlandsson Lampa, the County Administrative Board and the Swedish Water Authorities, for valuable discussions and review of the calculations with Method 3. Furthermore, we thank Babette Marklund, Järfälla Municipality, as well as Juha Salonsaari, Stina Thörnelöf, Jenny Pirard and Maria Pettersson, Stockholm City, for input and interesting discussions.

**Conflicts of Interest:** The authors declare no conflict of interest.

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
