# Peer review of "Methods for Calculation of Stormwater Treatment Required for Meeting Receiving Waters Quality: Application in a Case Study"

_sustainability, doi:10.3390/su142215395_

Round 1

Reviewer 1 Report

The Introduction section is not complete. Authors desribed background of research based only on 12 papers - it is not enough. In some paragraphs I couldn't find any citation. On page 3 lines 102-109 methods must be described. In this section I couldn't find novelty of proposed paper or method selection.

The methodology section was described properly. I suggest to described wider the pleca of case study - in the paper there is only few sentences about it.

Section 4 - Results and discussion - needs to be described wider. The Authors describe some informations about ibtained results but there is no comparison beetween used methods and results are not discussed with similiar works.

In the paper there is no Conclusions which must be added.

Reviewer 2 Report

This article (sustainability-1974195) entitled "Methods for Calculation of Required Treatment for Receiving Waters and Applied on a Case Study" presented a useful policy related and action-orientated methodology,  based on the need for tools on relating stormwater effluent data to the ambient water quality guidelines within the EU Water Framework Directive, after having performed sufficient literature review and found four operational tools (i.e., StormTac Web model, Vollenweider OECD load model, Swedish Water Authorities and SMHI model, and the method using Effluent concentration criteria) with compiled equations that can be used for calculation of the required annual treatment loads to work towards an ecological good status in the receiving waters. The topic is important for environmental science and policy making; the paper is very well constructed and written, which falls well within the scope of the journal. The paper is almost can be accepted in the present form. The only visible tiny issue is that many literature cited are written in Swedish. Hence, my suggestion is, citing the related illustration in the Supplementary material followed by a brief translation, which will pose important guidance for the related society.

Reviewer 3 Report

Many thanks for the invitation to review the manuscript.

I read with great interest your paper entitled “Methods for Calculation of Required Treatment for Receiving Waters and Applied on a Case Study”. This study aims to “to determine the loading capacity of the waterbody and to allocate that load among different pollutant sources so that the appropriate control actions can be taken, and water quality standards achieved.” The outcomes of this study provide the four methods applied to addressing total phosphorus in the urban watercourse Bällstaån, Sweden. Results showed good agreement of the calculated required treatment removals when measured phosphorus concentrations were used in Methods 1-2, but in other cases, there were larger deviations. The applied load method used in Method 3 that neglected the receiving water concentrations and criteria resulted in smaller required treatment removals. Method 4 also neglects the receiving waters concentrations and criteria, but nevertheless, it resulted in a similar calculated required treatment removal.

I would propose a “major revision” with the following comments:

1.      Abstract and conclusions need to be recondensed and re-written.

2.      The manuscript seems monotonous and simple to readers! Therefore, I suggest the authors should think how to improve the manuscript.

3.      The introduction does not fill the gap in literature of your research.

4.      The results and the discussion should be better highlighted. Particularly in the discussion, which method has the author's final decision recommended regarding the method's suitability for the research field?

5.      How can authors improve the research's limitations?

6.      For any study using models for simulation, it is important to include: the main equation(s); the key input data and the outpout (locations in the map of study area); and the sources of data; and how to calibrate/validate the models? Which parameters? Therefore, please insert these into your methodology session.

7.      Any standards for references must be included (maybe in Appendix).

Reviewer 4 Report

Please refer to the attached pdf

Round 2

Reviewer 1 Report

Thank You for comments and provided corrections. The paper is now properly.

Reviewer 2 Report

The authors have extensively modified the manuscript in the light of the suggestion provided by the reviewers.The overall quality of the manuscript has been further improved. Thus, I suggest acceptance of the present version of the manuscript for publication.

Reviewer 3 Report

The revised manuscript is accepted and recommended for publication.

Thank you.

Reviewer 4 Report

All comments for improvement has been addressed accordingly. The article is now well improved compared to the earlier version.